# Factors That Affect the Level of Success of the Transaction between Home Buyers and Developers in Sell-Build Residential Projects

**Selman Aslan** [1] , **David Arditi** [2,*] **and Gözde Tantekin-Çelik** [3]

1    Department of Transportation Services, Muş Alparslan University, Muş 49250, Turkey; se.aslan@alparslan.edu.tr

2    Department of Civil, Architectural, and Environmental Engineering, Illinois Institute of Technology, Chicago, IL 60616, USA

3    Department of Civil Engineering, Çukurova University, Adana 01330, Turkey; gtantekin@cu.edu.tr

*    Correspondence: arditi@iit.edu

**Abstract:** The "Sell-Build" model in residential project transactions involves a home buyer agreeing to pay the developer in monthly installments starting very early in the design or construction phases of a project. It is hypothesized that the success of this transaction depends on (1) the home buyer's knowledge about the transaction process and (2) the mutual trust between home buyer and developer. A survey was administered to 250 home buyers and 70 developers in Turkey to collect demographic information about the participants and about the participants' perceptions of not only the success of the transactions they were involved in, but also of the impacts of how informed home buyers are about the transaction process and how much the two parties trust each other. The data collected was subjected to statistical analysis. The findings indicate that (1) the home buyer should make a special effort to study relevant materials and consult specialists before entering into an agreement with a developer, and (2) the trust between home buyer and developer depends largely on the buyer's uninterrupted flow of monthly installment payments starting early in the project. The contribution of this study is that it provides a useful guideline to home buyers and developers.

**Keywords:** construction industry; residential projects in Turkey; trust; knowledge; transaction success

## 1. Introduction

Residential construction projects make an important contribution to economic growth, employment opportunities, and the general wellbeing of a nation, as they satisfy human beings' basic need of shelter (Olayiwola et al., as cited in [1]). Currently, given the increased growth in the urban population and the improved quality of life, residential projects must meet different personal expectations as well as the basic needs of shelter and protection. According to Forsythe [2], the residential construction sector will adjust itself to meet both the objective and subjective expectations of buyers. For example, to meet the extensive housing demand, many developing countries encourage the development of public residential projects to help low-income citizens to become homeowners and to live at an acceptable standard at a reasonable price (Ibem and Azuh, as cited in [1]).

Residential projects can be developed using various models such as "turnkey", "build-sell", and "sell-build". The "sell-build" model that involves a developer building the residential units only after the units are sold to buyers is quite common as it enhances the home buyer's purchasing power by allowing the home buyer to pay by monthly installments. The "sell-build" model also enhances the cash flow of the developer's operation since home buyers pro-actively help to finance the project by making regular monthly payments during design and construction [3]. In the "sell-build" model, the developer may be a contractor that does some of the work and subcontracts the rest, or simply an investor who coordinates the works of several contractors.

In this study, it is hypothesized that (1) the extent to which the home buyer is informed about the transaction, and (2) the trust between the home buyer and the developer has an impact on the success of the transaction between the home buyer and the developer. The effects of "buyer's knowledge about the transaction" and "trust between developer and buyer" on transaction success are investigated. Developers and buyers are surveyed, and the collected data are statistically analyzed. The results are expected to guide developers and home buyers in future transactions. The success/failure of these transactions is important not only for developers and home buyers, but also for national development [4]. The determination of the conditions that are conducive to a successful transaction between developer and home buyer can serve as a guideline in future projects and can allow the parties to eliminate the deficiencies. Indeed, it can put an end to the grievances of home buyers who are totally dissatisfied with a sell-build venture after committing all their savings to the venture as many middle-class individuals do. It can also bring clarity to good practices to be pursued by developers. Consequently, a healthy continuity can be ensured in the development of residential projects.

## 2. The Residential Construction Industry

The residential construction industry is important not only for individuals to reach prosperity and for developers to flourish, but also for the national economy to grow. The construction industry is vital to the development of a nation because the physical development of construction projects is a measure of economic growth [5]. A growing economy depends on the development of a country's physical infrastructures such as roads, bridges, coastal structures, power plants, and residential and non-residential buildings [6]. The construction industry has extensive ties with many other industries and acts as a catalyst for growth in industries such as manufacturing, transportation, power generation, and financial services [7].

Residential projects are an important part of the construction industry. The existing housing stock is always in need of restocking [8]. For example, despite the significant increase in the number of residential units in Turkey in recent years, the need for quality housing construction is on the rise because of demographic reasons especially in urban areas [9]. Residential units are marketed by developers by using various sales strategies such as presenting a three-dimensional model to the potential buyer or arranging a visit to a fully built model of a sample unit. While developers focus on delivering the project on schedule, within budget, and in line with the home buyer's needs, buyers are concerned about value for money, total completion without delay, and quality of the residential unit [10].

The evaluation and decision of the home buyer involve six steps: identifying the need, searching for alternative projects, evaluating the alternatives, purchasing, evaluating the performance of the constructed facility, and judging the level of satisfaction with the transaction [2]. While the first three steps of the process affect the pre-purchase expectations of the home buyer, the last three steps involve the home buyer's perceptions of the outcomes of the transaction between the home buyer and the developer.

### 2.1. Studies on Transaction Success

The definition and classification of project processes and success factors vary from sector to sector. Generally, a construction project is considered successful when completed on schedule, within budget, in compliance with specifications, and in a way that receives stakeholder satisfaction [11,12]. Construction projects are particularly affected by the environment in which they are built. The success of most construction projects is also heavily influenced by the physical, political, and social environment, cultural traditions, and especially human-induced factors that differ from country to country [4].

The developer is the party who manages the quality of a project. In the "sell-build" delivery method, once the purchase agreement is signed between the developer and the buyer, construction is under the control of the developer who makes sure that the project is undertaken in accordance with the specifications agreed to with the buyer [5]. The success

of the developer in meeting quality requirements depend on project complexity, and the developer's technical expertise, organizational competence, track record, financial stability, and risk management practices [4,5,13].

Even though some researchers explored the effect of different criteria, the success criteria that are used to measure the success or failure of a real estate transaction are mostly based on the cost, time, and quality of the constructed facility. While using the objective criteria of time, cost, and quality, it should be noted that expectations may differ from country to country depending on the work environment [7].

### 2.2. The Trust between the Developer and the Home Buyer

The "sell-build" delivery system is preferred by many home buyers because it allows easy payment by installments as opposed to full upfront payment. Home ownership is higher in some countries than others but is a common desire in most countries as it symbolizes long-term security and is an investment that eliminates rental payments and eases the eventual transfer of the property to heirs. Potential homeowners want to acquire a residential property as soon as possible in their lives. People with fixed income who do not have enough readily available cash to buy a house tend to purchase homes using the sell-build system that allows them to pay in small installments. The "sell-build" delivery system is also preferred by many developers because it provides upfront financing during design and construction, and because it allows the developer to manage the operation's cash flow with minimum borrowing. In the sell-build system, the home buyer develops a sense of trust based on the suitability of the payment method, the duration of the work, the competence of the builder, the quality of the work done, and accessibility to the developer. The developer, on the other hand, develops a sense of trust based on the buyer's consistent and timely installment payments, the buyer's genuine appreciation of the work done, effective communications with the buyer, and effective mutual problem-solving behavior.

When the studies about "trust" between the stakeholders of a construction project are examined, it is seen that trust is affected by many factors, is defined by many conceptual classifications, and is measured by different survey tools. For example:

- Trust can be conceptualized in the construction industry by considering system-based trust (organizational policy, communication system, and contracts/agreements), cognition-based trust (communication and knowledge), and impact-based trust (emotional investments and thinking) [14]. Trust can also be categorized as reckoning trust (logical), relational trust (individual relations), and institutional trust (organizational regulations) [15].
- The characteristics that affect the trust level of partners in a cooperative system include competence in the work performed, problem solving capability, effectiveness of communication, openness, alignment of effort and rewards, flow of knowledge, sense of unity, respect and appreciation of the system, compliance, long-term relationships, financial stability, reputation, adoption of alternative dispute resolution techniques, and clarity in contracts [16]. These characteristics can also be addressed by eight factors, namely history of interaction (the most important factor), knowledge sharing and communication, contracts and institutions, competence, relationship-specific investment, reputation, honesty, and opportunistic behavior [17].
- The independent variables affecting trust were identified as reputation, competence, integrity, communication, reciprocity, and contract [18]. On the other hand, when trust between the stakeholders of a project is investigated, reliable behavior, good communication, sincerity, competence, integrity, achieving project milestones, commitment, benevolence, and purpose-harmony come into play [19].
- Teamwork and communication affect participants' behaviors and general project knowledge, so that trust-based cooperation is more likely to emerge and continue [20]. A trust-based relationship between the contracting parties could achieve a better risk management process, which could also reduce cost [21]. According to Khalfan et al. [22], trust between suppliers to construction projects is based on basic factors

such as experience, problem solving, common goals, reciprocity, and reasonable behavior.

To summarize, the studies mentioned in these items directly describe the many factors that affect trust, but from different perspectives.

### 2.3. The Home Buyer's Knowledge of the Transaction

In residential projects based on the sell-build system, the home buyer is expected to have enough knowledge about residential projects and the purchasing process, and yet in general, home buyers do not have a good understanding of the housing market (Arıkan, as cited in [23]) because most buyers are lay people (i.e., they are not informed or knowledgeable buyers). They may have different occupations (e.g., professional, housewife, student, retired, civil servant, etc.), may be in different age groups (i.e., young, middle aged, old), and may have different educational backgrounds (grade school, high school, college, advanced degree, etc.), but many buyers may be first-time buyers with no prior experience in the housing market.

The home buyer signs a contract with the developer when purchasing a residential unit. In principle, this contract should cover all the processes involved in the residential project and regulate the relationship between the home buyer and the developer. For example, the contract should state how disagreements and disputes will be resolved. The contract should state how any delays and additional costs will be handled, and how quality-related issues will be resolved. It is to the home buyer's advantage to read and understand all the articles of the contract because caveat emptor (i.e., "let the buyer beware") is a universal principle embedded in commercial law that stipulates that notwithstanding the plans, specifications, and warranties implied in a contract, the buyer purchases a property at his/her own risk.

In addition, before agreeing to sign a contract with the developer, the buyer needs to investigate the developer's performance in past projects and assess the level of satisfaction of past customers; the buyer needs to be informed about the economic situation in general (i.e., the inflation rate, credit policies, etc.); and the buyer needs to thoroughly study the terms of the contract (including the general responsibilities of the developers, the rights of the buyer, and the processes involved in managing the project) [24,25]. For example, an important criterion for a home buyer may be the number of rooms and sun exposure, while for another buyer, it may be the floor area and the neighborhood where the property is located. Home buyers should know the features of the dwelling they want to purchase and should be able to define them briefly and clearly [26]. Lundgren and Lic's [27] research reveals that home buyers' preferences are sometimes related to abstract values (such as psychological considerations and entrenched belief systems) even though there is evidence in Mary and Surulivel's [28] study that for most home buyers, quality, price, location, and delivery time are the most important factors that affect buyer satisfaction. In a survey conducted by Ratchatakulpat et al. [29] of 376 potential home buyers in Australia, price, usage, interior design, and the neighborhood were determined as the most important factors in selecting a residence. However, according to Źróbek et al. [30], there may be additional reasons that affect a home buyer's decision in housing selection, including factors such as security, proximity to schools or parks, and distance to the city center. Źróbek et al. [30] conclude that in Poland, the price of the home, quiet neighbors, and safety are the three most important factors.

In Turkey, the guidance issued by the Association of Real Estate Investment Association includes answers to questions such as "Have you researched the developer of the residential unit you are considering buying?", "Do you have enough knowledge about the purchasing process and of your rights in this process?" [31]. In a different booklet of the same association, home buyers are advised to acquire the necessary knowledge about how to purchase a residential unit before going ahead with the purchase [32]. Such guidelines are issued as a service to first-time home buyers or buyers who are not well informed about this process.

## 3. Research Methodology

A questionnaire was developed for this study that investigates the sale of residential units before the start of construction or during construction. The study covered residential projects that were being designed, under construction, or sold in Istanbul, Ankara, and Izmir, the three largest provinces in Turkey. The potential respondents associated with these projects were approached at random. Interviews were held with 250 home buyers who had just purchased residential units in the design phase or during construction in the last two years and 70 developers who built and sold these units. The information was collected in November 2019 using a mixed survey design that included email communications, telephone calls, and face-to-face interviews. Each interview lasted an average of 20 min.

The first set of survey questions sought general demographic and descriptive information about the characteristics of the respondents. The survey also included a set of questions about the buyers' knowledge of the purchasing process, and another set about the trust between developer and buyer. The respondents were also asked about their perception of the success of the transaction.

As seen in Table 1, five hypotheses were developed to test the effects of (1) home buyers' knowledge of the purchasing process, and (2) the trust between the home buyer and the developer on transaction success.

**Table 1.** Modeling hypotheses.

| No | Hypotheses |
|----|-----------|
| 1 | The more knowledgeable the home buyer is about the purchasing process, the higher the success of the transaction. |
| 2 | The more trust the home buyer has in the developer, the higher the success of the transaction. |
| 3 | The more trust the developer has in the home buyer, the higher the success of the transaction. |
| 4 | The developer's perception of the home buyer's knowledge of the purchasing process does not agree with the home buyer's actual knowledge of the transaction process. |
| 5 | The developer's trust in the home buyer is not related to the home buyer's trust in the developer. |

### 3.1. Home Buyers' Knowledge about the Purchasing Process of Residential Units

The 20 questions about the home buyer's knowledge of the home purchasing process were answered on a 5-point Likert-type scale. Likert-type questions are categorical and are prepared in sequential options from lowest to highest or vice versa. The questions were prepared by using the 13 knowledge items identified in the studies presented in Table 2. The scope of some of the knowledge items was wide enough to require representation by several questions, which explains why as many as 20 questions were formulated to represent only 13 knowledge items. While this part of the questionnaire was administered to home buyers, a reworded version was administered to developers to seek developers' perceptions of home buyers' knowledge about the home purchasing process. The knowledge of the purchasing process reported by home buyers was then compared to developers' perceptions of the same items.

**Table 2.** Knowledge items used by home buyers.

| Knowledge Items | Researchers |
|---|---|
| Financial stability of the developer Developer's ownership of adequate resources | Dang et al. (2012) [4], Alzahrani and Emsley (2013) [5], Yong and Mustaffa (2013) [7], Belassi and Tukel (1996) [33], Kuwaiti et al. (2018) [34] |
| Project management skills of the developer | Alzahrani and Emsley (2013) [5], Kıvrak et al. (2008) [24], Kuwaiti et al. (2018) [34], Kanapeckiene et al. (2010) [35], Nilashi et al. (2015) [36] |
| Contents of and fairness in the contract | Forsythe (2007) [2], Dang et al. (2012) [4], Yong and Mustaffa (2013) [7], Nguyen et al. (2004) [12], Iyer and Jha (2006) [37], Toor and Ogunlana (2008) [38], Rashvand and Majid (2014) [39] |
| Value for money invested in the transaction Prediction of the future value of the property | Chan et al. (2002) [11], Zou et al. (2006) [26], Rashvand and Majid (2014) [39], Coff (1999) [40], Ling and Bui (2010) [41], Xiang et al. (2013) [42] |
| Socioeconomic factors | Alzahrani and Emsley (2013) [5], Nilashi et al. (2015) [36], Iyer and Jha (2006) [37], Ahadzie et al. (2007) [43] |
| Performance in past projects | Dang et al. (2012) [4], Alzahrani and Emsley (2013) [5], Doloi et al. (2010) [13], Bryde and Robinson (2005) [44] |
| Developer's appreciation for cost-time-quality issues | Zou et al. (2006) [26], Zou et al. (2007) [45] |
| Legal responsibility for contractual commitments | Kariya et al. (2016) [46] |
| Type of project undertaken | Koklic and Vida (2011) [25] |
| Awareness of the property's surroundings | Lundgren and Lic (2009) [27] |
| Appreciation for the environment | Ratchatakulpat et al. (2009) [29], Źróbek et al. (2015) [30], Zou et al. (2007) [45] |
| Other buyers' opinions | Mary and Surulivel (2014) [28], Źróbek et al. (2015) [30] |
| Reliance on technical/legal consultants | Koklic and Vida (2011) [25], Argan (2012) [47] |

### 3.2. Trust between Developer and Buyer

After reviewing many academic studies that have been performed to investigate trust between parties, the questionnaire developed by Tai et al. [17] was adopted in this study because this questionnaire covers all the concepts mentioned in the literature review and used in all survey tools combined. The 24 questions about trust items are presented in Table 3. One set of questions was about the home buyer's trust in the developer. The questions were coded on a Likert scale of 1–5. This survey (see Appendix A) was administered to home buyers to measure the trust of home buyers in developers. A reworded version of the same survey was administered to developers to measure developers' trust in the home buyers (see Appendix B).

**Table 3.** Items of trust between home buyer and developer [17].

| Trust Items | Short Descriptions |
|---|---|
| Competence | Skills and qualifications |
| Honesty | Integrity, morality, authenticity |
| Problem solving mechanism | Settlement of disagreements and dispute resolution |
| Similarity | Joint values |
| Information sharing | Transparency and access to information when needed |
| Keeping promises made | Delivering commitments as specified in the contract |
| Reputation | Professional standing and prestige |
| Mutual respect | Acknowledgment of the other party's interests and wishes |
| Long-term cooperation | Track record of long-term relationship |
| Fairness | Fairness in decision-making, equity |
| Effective communication | Timely and unambiguous exchange of information |
| Frequent communication | Frequent exchange of information |
| Consistency between efforts and rewards | Mutual understanding of the consequences of actions taken by either party |
| Predictability of parties' expected behavior | Expectation of reliable and consistent behavior |
| Confidence in the other | Reasonable belief in the other party's legal and ethical performance |
| Completeness of contract | Thorough, explicit, and well thought out contract |
| Presence/absence of opportunistic behavior | Opportunistic behavior may affect the transaction to the detriment of the other party |
| Supervision by third party | Supervision of the developer by the home buyer's technical/legal consultant; supervision of the home buyer by the developer's financial/accounting consultant |
| Level of effort spent to achieve goals | Parties' clear commitment to achieving mutual goals |
| Level of interaction | Mutual and consistent engagement and exchange of information between the parties |
| Sense of social responsibility | Consideration given to socio-economic considerations |
| Good intentions | Commitment to not doing harm to the other party |
| Common goals | Unity of purpose |
| Productive interdependence | Understanding that one party's well-being/ satisfaction is closely related to the other party's well-being/satisfaction |

## 4. Transaction Success

As per the consensus in the literature (see Section 2.1), transaction success was measured by project actual cost vs. initially budgeted, actual completion time vs. initially scheduled, and actual quality vs. initially specified. As seen in Equation (1), the overall success of the transaction is the weighted average of the three success factors and was

calculated by multiplying the score in each of the factors by the respective weight assigned by the respondents.

$$OST = [(Sc \times Wc) + (St \times Wt) + (Sq \times Wq)] / (Wc + Wt + Wq) \qquad (1)$$

where OST represents the overall success of the transaction, $Sc$, $St$, and $Sq$ represent the scores for cost, time, and quality, while $Wc$, $Wt$, and $Wq$ represent the weights specified by the respondents for cost, time, and quality.

SPSS 20.0 and Microsoft Excel 2020 were used to analyze the data collected. The Kolmogorov–Smirnov significance level and the kurtosis–skewness coefficients were examined for the compatibility of the data to a normal distribution. The Mann–Whitney U Test was used to examine the effects of the participants' demographic characteristics on the success of the transaction. Exploratory factor analysis was performed to reduce the large numbers of the factors that measure home buyers' knowledge about the purchasing process and the trust between home buyer and developer.

## 5. Findings

The demographic characteristics of the 250 home buyers who participated in the survey are summarized in Table 4. The following can be observed from this information.

- Male buyers represent the majority of the respondents with 68% of all buyers, while female buyers constitute only 32% of buyers, an expected result in a patriarchal society.
- If one considers the age distribution of the buyers, mostly young and middle-aged individuals in the 25–39 age group (82% of the participants) buy residential units from developers, which implies that the older generations are already homeowners.
- The education level of the participants is generally high with 61% having a college degree, 38% an associate degree or a high school diploma, and only 1% less education, which ties in with the participants' occupations.
- Most buyers are civil servants, workers, and technical personnel (78%), indicating that salaried (fixed income) individuals tend to buy residential units more than managers, self-employed, and unemployed individuals (22%).
- Buyers of all income levels purchase residential units from developers.

The respondents were asked to mark the importance of the three success factors (i.e., actual cost vs. budgeted cost, actual duration vs. scheduled duration, actual quality vs. quality specified in the contract) that affect the success of the transaction in the purchase of a residential unit as well as their perception of the actual success of the transaction. The responses are shown in the bottom six rows of Table 3. Using Equation (1), the overall success of the transaction was calculated according to the weights of cost, time, and quality as specified by the respondents. Concerning the characteristics of the developers, their education level is generally high with 56% having a college degree, 36% an associate degree or a high school diploma, and 9% less education.

The Cronbach's Alpha coefficients calculated for the knowledge and trust constructs are shown in Table 5. For a construct to be reliable, the Cronbach's Alpha coefficient must be at least 0.7 (Nunnaly, as cited in [48]). Three expressions (14th, 15th, and 18th statements in Appendices A–C) were removed from the knowledge construct to improve the reliability of the construct, reducing it to 17 statements. As seen in Table 5, all Cronbach's Alpha coefficients are above the threshold of 0.7 after this revision is made.

Home buyers' knowledge of the purchasing process was found by calculating the average of the responses to 17 statements in the revised construct. The level of trust between developers and home buyers was found by calculating the average of the responses to 24 expressions in the trust construct. The means of the responses for each statement are presented in Appendix C using the 1–5 Likert scale, where 1 represents "strongly disagree" and 5 "strongly agree".

**Table 4.** Distribution of responding home buyers by their demographic characteristics.

| Information about Home Buyers | | Number of Respondents (n) | Percentage of Respondents (%) |
|---|---|---|---|
| Gender | Female | 80 | 32 |
| | Male | 170 | 68 |
| Age | 25–29 years old | 71 | 28 |
| | 30–34 years old | 73 | 29 |
| | 35–39 years old | 63 | 25 |
| | 40–44 years old | 25 | 10 |
| | 45 years and older | 18 | 7 |
| Education status | Middle school diploma | 2 | 1 |
| | High school diploma | 56 | 22 |
| | Associate degree | 39 | 16 |
| | Undergraduate degree | 126 | 50 |
| | Graduate degree | 27 | 11 |
| Occupation | Unemployed | 18 | 7 |
| | Self-Employed | 20 | 8 |
| | Manager | 18 | 7 |
| | Technical personnel | 43 | 17 |
| | Civil servant | 79 | 32 |
| | Worker | 72 | 29 |
| Monthly income | Between 2500–4000 TL | 38 | 15 |
| | Between 4001–5500 TL | 74 | 30 |
| | Between 5001–7000 TL | 69 | 28 |
| | Between 7001–8500 TL | 33 | 13 |
| | Between 8501–10,000 TL | 13 | 5 |
| | Over 10,000 TL | 20 | 8 |
| | Declined to answer | 3 | 1 |
| The most important selection criterion | Duration | 59 | 24 |
| | Cost | 59 | 24 |
| | Quality | 132 | 53 |
| The second most important selection criterion | Duration | 79 | 32 |
| | Cost | 111 | 44 |
| | Quality | 60 | 24 |
| The third most important selection criterion | Duration | 112 | 45 |
| | Cost | 80 | 32 |
| | Quality | 58 | 23 |
| Success of the transaction (actual duration vs. scheduled duration) | Successful | 221 | 88 |
| | Unsuccessful | 29 | 12 |
| Success of the transaction (actual cost vs. budgeted cost) | Successful | 222 | 89 |
| | Unsuccessful | 28 | 11 |
| Success of the transaction (actual quality vs. quality specified in the contract) | Successful | 206 | 82 |
| | Unsuccessful | 44 | 18 |
| **Total** | | **250** | **100** |

**Table 5.** Reliability coefficients.

| Sample | Scale | Number of Statements | Cronbach's Alpha Coefficient | Revised Number of Statements | Revised Cronbach's Alpha Coefficient |
|---|---|---|---|---|---|
| Home buyer | Knowledge | 20 | 0.661 | 17 | 0.704 |
| | Trust | 24 | 0.784 | 24 | 0.784 |
| Developer | Knowledge | 20 | 0.822 | 17 | 0.814 |
| | Trust | 24 | 0.921 | 24 | 0.921 |

The 17-statement revised knowledge construct and the 24-statement trust construct do not fit the normal distribution because the Kolmogorov–Smirnov significance level was less than 0.05 for all data and some of the kurtosis–skewness coefficients were outside the range [−2.00, +2.00]. Therefore, the nonparametric Mann–Whitney U test was performed to test the hypotheses mentioned in Table 1. The Mann–Whitney U test results are presented in Table 6 and reveal the following.

**Table 6.** Mann–Whitney U test results.

| No | Relationships Reviewed | p-Values | Significance Level | Statistical Significance |
|---|---|---|---|---|
| 1 | The buyer's knowledge of the purchasing process vs. transaction success | 0.032 | $p < 0.05$ (Significant) | Yes |
| 2 | The buyer's trust in the developer vs. transaction success | 0.007 | $p < 0.01$ (Highly Significant) | Yes |
| 3 | The developer's trust in the buyer vs. transaction success | 0.021 | $p < 0.05$ (Significant) | Yes |
| 4 | The developer's perception of the home buyer's knowledge of the purchasing process does not agree with the home buyer's actual knowledge of the transaction process. | 0.000 | $p < 0.001$ (Very Highly Significant) | Yes |
| 5 | The developer's trust in the home buyer is not related to the home buyer's trust in the developer. | 0.059 | $p > 0.05$ (Not Significant) | No |

1. Hypothesis 1 holds: There is a significant relationship between the home buyer's knowledge of the purchasing process and the success of the transaction. The transaction is perceived to be successful when home buyers are well informed about the purchasing process, confirming the general belief in the relevant literature that sound knowledge about the purchasing process is of great importance in real estate transactions [24,35,49]. Those buyers without adequate knowledge of construction and the purchasing process are likely to be involved in transactions that may not be successful in at least one of the success measures of time, cost, or quality. It is, therefore, important that the home buyers make a special effort to get informed about basic construction and the purchasing process. Particularly home buyers who buy for investment purposes have extensive knowledge of the transaction process [50].
2. Hypothesis 2 holds: There is a significant relationship between the buyer's trust in the developer and the success of the transaction. The transaction is perceived to be successful when home buyers trust developers implicitly. This result is confirmed by several studies (e.g., Wong et al. [14]; Tai et al. [17]; Jiang et al. [18]; Karlsen et al. [19]; Kadefors [20]; Zaghloul and Hartman [21]; Bas Aras [51]; Laan et al. [52]). When

purchasing a home, the buyer makes a choice between several alternatives. The home buyer's trust in the developer's honesty and integrity greatly influences this choice. It is, therefore, important that the developer acquires a track record of honesty, smooth interaction with home buyers, sincerity, risk sharing, unity of purpose with home buyers, and good communications.

3. Hypothesis 3 holds: There is a significant relationship between the developer's trust in the home buyer and the success of the transaction. The transaction is perceived to be successful when developers trust home buyers implicitly. Wong et al. [14], Tai et al. [17], Jiang et al. [18], Karlsen et al. [19], Kadefors [20], Zaghloul and Hartman [21], Laan et al. [52], and Chan et al. [53] investigated the importance of mutual trust between the developer and the home buyer. They found that the developer relies on a trustworthy home buyer for a smooth transaction, i.e., a home buyer who makes regular payments, who reacts sensibly to the inevitable problems that may arise in the construction process, and who praises the developer to future potential buyers. For example, Ling and Bui [41] report on cases that involve home buyers who stop their monthly installments in a residential project that is behind schedule and is progressing too slowly due to design changes and technical problems, which consequently causes the project to fail.

4. Hypothesis 4 holds: There is a significant difference between the home buyer's knowledge of the purchasing process and the developer's perception of this knowledge, indicating a serious mismatch in the opinions of the two parties. It is the opinion of developers that home buyers' perceived knowledge of the purchasing process is unrealistically inflated, as evidenced by the higher average of the home buyers' responses than the average of the developers' responses. It can be inferred that the level of home buyers' knowledge about the purchasing process is not as realistic as the buyers thought and that buyers who are better informed can enhance their understanding of the marketplace. On the other hand, the finding relative to Hypothesis 4 may also imply that home buyers are better informed than developers' observations of the buyers' knowledge of the transaction. It may also mean that the developers underestimate the home buyers' real knowledge. In both cases, the inference that buyers who are better informed can enhance their understanding of the marketplace does not change, but it reflects a more favorable condition for home buyers.

5. Hypothesis 5 does not hold: There is no significant difference between the developers' trust in home buyers and the home buyers' trust in developers. This result reinforces the findings of the tests of Hypotheses 2 and 3 that mutual trust between developer and home buyer has a significant effect on the success of the transaction.

### 5.1. Exploratory Factor Analysis of the Construct That Measures the Buyer's Knowledge of the Purchasing Process

Factor analysis reduces very many variables into fewer factors by bringing together the variables that are correlated with each other. It is easier to interpret and explain fewer factors representing several variables. The sample size of 250 home buyers was large enough to perform factor analysis since it exceeds the 100-threshold recommended by Hair et al. [4]. However, since the sample of 70 developers was below the 100-threshold recommended by Hair et al. [54], factor analysis could not be used in this group. The construct that measures the buyer's knowledge about the purchasing process consists of 17 statements. In the sample of 250 home buyers, as per Hair et al.'s recommendation, factor loadings of 0.35 and above are considered significant [55]. Two items overlapping with other items were excluded from the analysis, reducing the number of items to 15. As recommended by Hair et al. [54], the Kaiser–Meyer–Olkin (KMO) and the Bartlett Sphericity tests were conducted to check for suitability for factor analysis. Indeed, the Kaiser–Meyer–Olkin measure of sampling adequacy exceeded 0.50. Moreover, the KMO value (0.725) was found to be good and the Bartlett's Test of Sphericity ($x2 = 728.338$; $p = 0.00$) was statistically significant. The final version of the factor structure obtained after applying Varimax rotation is presented in Table 7, which shows that the construct

that measures buyers' knowledge about the purchasing process can be represented by six factors that explain a little over 66% of the total variance. The contributions of individual factors to the total variance are 21% for the first factor; 13% for the second factor; 9% for the third factor; 9% for the fourth factor; 7% for the fifth factor, and 7% for the sixth factor. The six factors that represent this construct are named knowledge about the terms of the contract, the developer's past and present performance, the developer's competence, purchasing strategy, legal matters, and market research.

**Table 7.** Factor pattern of the knowledge construct.

| Items | Factors | | | | | |
|---|---|---|---|---|---|---|
| | **1** | **2** | **3** | **4** | **5** | **6** |
| I made a detailed comparison between the information specified in the blueprints and the information specified in the contract. | **0.792** | 0.099 | 0.032 | 0.113 | 0.118 | 0.121 |
| I am familiar with the brands, standards, and quality of materials used in the construction. | **0.747** | 0.228 | 0.035 | −0.084 | −0.132 | 0.000 |
| The contract I signed contains all the construction related details I expect to see in such a contract. | **0.712** | −0.061 | 0.312 | −0.030 | 0.087 | −0.062 |
| I know how much my home will cost at handover. | **0.658** | 0.465 | −0.045 | 0.104 | 0.058 | −0.101 |
| I know about the level of success and reputation of the developer in past projects. | 0.145 | **0.819** | −0.007 | −0.050 | 0.014 | −0.087 |
| After handover, I know how long it took the developer to build the superstructure and to complete the specialty work. | 0.030 | **0.782** | 0.075 | 0.024 | 0.013 | 0.199 |
| I can calculate the future value of the current money I invested in the building. | 0.239 | **0.728** | 0.057 | −0.030 | −0.183 | −0.080 |
| I researched the developer's finances and human/equipment resources. | 0.083 | 0.049 | **0.867** | 0.001 | 0.113 | 0.019 |
| I know the developer's project management capabilities. | 0.129 | 0.060 | **0.821** | 0.124 | 0.056 | 0.091 |
| I made the decision to purchase my home rationally rather than emotionally. | −0.050 | 0.014 | 0.082 | **0.807** | 0.106 | −0.148 |
| I researched the proposed landscaping around my potential home and the potential development of the neighborhood. | 0.102 | −0.033 | 0.022 | **0.767** | −0.148 | 0.183 |
| I am familiar with dispute resolution methods. | 0.063 | −0.050 | 0.038 | −0.191 | **0.795** | −0.052 |
| I know construction terms such as title deed, property, easement, mortgage, and building permit. | 0.030 | −0.126 | 0.213 | 0.336 | **0.600** | 0.079 |
| I talked to other buyers about the residential development project. | 0.001 | −0.117 | 0.081 | 0.063 | −0.133 | **0.816** |
| I compared the residential development project with alternative projects. | 0.004 | 0.237 | 0.028 | −0.059 | 0.399 | **0.591** |

Bolded numbers represent significant factor loadings.

*Factor 1: Knowledge about the Terms of the Contract*

It is extremely important for the home buyer to read and understand the contract between the developer and the home buyer. This factor's contribution to total variance is larger than any other factor's contribution. This finding is supported by many studies (e.g., Dang et al. [4]; Nguyen et al. [12]; Wong and Cheung [16]; Tai et al. [17]; Jiang et al. [18]; Kadefors [20]).

*Factor 2: Knowledge about the Developer's Past and Present Performance*

The contract between developer and buyer is terminated at handover, but in Turkey, the responsibility of the developer continues for fifteen years for defects related to the workmanship and materials used in the load-bearing system of the building, and for two years for other non-load-bearing parts [55]. The performance of the developer is tested not only in the design and construction phases of the project, but also in the warranty period. In Malaysia, the warranty period is between 18 and 24 months after handout [46]. In a study conducted by Baş Aras [51], it was observed that statements such as "we care about customer satisfaction after sales" came to the fore in housing sales. In addition, Dang et al. [4], Alzahrani and Emslev [5], Doloi et al. [13] and Bryde and Robinson [44] have identified developer performance as a major issue considered by most home buyers. Finally, a home buyer who has enough knowledge of engineering economics principles (i.e., the time value of money) and who can calculate the future value of the installment payments made to the developer, can also make an informed assessment of the financial soundness/solvency of the developer. This level of sophistication is a distinct advantage in the sell-build environment where the home buyer is in a long-term relationship with the developer. Indeed, if the developer goes bankrupt in this process, the home buyer risks losing all the hard-earned money invested in this project. It is, therefore, important for a home buyer to have a good idea of whether the developer's finances are sound and stable.

*Factor 3: Knowledge about the Developer's Competence*

The financial condition and the project management capabilities of the developer are of great importance in transaction success and have a direct bearing on the buyer's decision to purchase. Knowing that the developer is financially sound and has the necessary managerial capabilities reassures the buyer that the project will not be discontinued. Indeed, many studies found that solid finances and competence are conducive to project or transaction success (e.g., Dang et al. [4]; Nguyen et al. [12]; Doloi et al. [13]; Kuwaiti et al. [34]; Toor and Ogunlana [38]; Rashvand and Majid [39]).

*Factor 4: Knowledge about Purchasing Strategy*

Some buyers make a rushed decision to purchase a home with the excitement of owning a home, without thinking over the consequences thoroughly. It is known that psychological factors do influence people's choice of housing [27]. The buyers who decide to buy a home by thinking strategically research the residential project and the potential development around the building's neighborhood in a systematic way because location has a significant direct effect on bargaining and an indirect effect on the value of the house [56]. As a result, they are in a better position to benefit from this purchase. Buyers who make rational rather than emotional purchasing decisions are likely to be involved in successful transactions with developers.

*Factor 5: Knowledge about Legal Matters*

Conflicts may arise between the home buyer and the developer for reasons such as cost overruns, delays, contract breaches, and problems with quality. It is in the buyer's interest to understand the contract language. The buyer should also know how potential disputes can be avoided and how they can be resolved if they occur [16,26,39].

*Factor 6: Knowledge about Market Conditions*

The home buyer typically considers multiple residential project alternatives. There are also several parameters that characterize a home's relative attractiveness and determine its price [57]. So, the residential unit selected by the buyer must meet the buyer's needs and expectations. It helps if the buyer exchanges ideas with other buyers of homes in multiple residential projects [2,34,43].

Overall, it can be stated that home buyers must be well informed about the six factors identified in factor analysis, but especially about the factor that has the largest factor loadings, namely the terms of the contract. Home buyers should particularly be well informed about the responsibilities and liabilities of the parties as specified in the contract; payment routines, including the timing of the payments, the handling of cost escalation, late payments, and penalties; time-related issues including progress reports and the handling of delays; settlement of claims and resolution of disputes if any; monitoring of the quality of construction and remedies for defective work; and other issues specified in the contract.

*5.2. Exploratory Factor Analysis of the Construct That Measures the Buyer's Trust in the Developer*

The construct that measures trust consists of 24 statements. Two statements overlapping with other statements were removed, reducing the number of statements to 22. The KMO value (0.784) was good and the Bartlett's Test of Sphericity (x2 = 1659.557; $p$ = 0.00) was statistically significant. The final version of the factor structure presented in Table 8 was obtained after applying Varimax rotation. As a result of the factor analysis, the trust construct was represented by six factors, which explain 61% of the total variance. The contributions of the individual factors to the total variance were 19% for the first factor; 16% for the second factor; 8% for the third factor; 6% for the fourth factor; 5% for the fifth factor, and 5% for the sixth factor. The six factors were named integrity, close interaction, harmony, fair play, unity of purpose, and effective communication.

*Factor 1: Integrity*

The buyer believes the developer has integrity if the developer does not take advantage of the buyer, performs reliably, proposes a contract that is clear and fair, and is truthful about his/her competencies and resources. Integrity/honesty is often mentioned as an important dimension of trust in several studies (e.g., Wong and Cheung [16]; Tai et al. [17]; Jiang et al. [18]; Karlsen et al. [19]; Khalfan et al. [22]).

*Factor 2: Close Interaction*

The home buyer and the developer are in a long-term relationship covering the design and construction phases of a project. Constant interaction is desirable between the home buyer and the developer and involves sharing the same values, sharing information, and solving problems by means of friendly negotiations. The developer's positive reputation constitutes evidence of a smooth interaction between the home buyer and the developer. This factor is also named "reciprocity" in some studies (e.g., Wong and Cheung [16]; Tai et al. [17]; Jiang et al. [18]; Khalfan et al. [22]).

*Factor 3: Harmony*

Close and sincere engagement with each other, mutually displaying social responsibility, publicly recognizing each other's contributions to the project, and protecting each other's interests are evidence of a harmonious relationship between the home buyer and the developer. Sincerity is essential in a harmonious relationship and was emphasized in a study by Karlsen et al. [19] as an important factor that affects the trust between home buyers and developers.

**Table 8.** Factor pattern of the trust construct.

| Items | Factors | | | | | |
|---|---|---|---|---|---|---|
| | **1** | **2** | **3** | **4** | **5** | **6** |
| The developer does not take advantage of the weak points in the contract. | **0.795** | −0.062 | 0.070 | 0.012 | −0.086 | 0.031 |
| The developer acts reliably and can meet the buyer's expectations. | **0.785** | 0.022 | −0.049 | −0.002 | 0.091 | 0.048 |
| The rights and obligations of the developer and the buyer are clearly expressed in the contract. | **0.767** | −0.091 | 0.147 | 0.076 | −0.006 | 0.043 |
| The buyer has confidence in the competence of the developer. | **0.765** | 0.023 | 0.124 | −0.036 | −0.005 | −0.100 |
| A consulting firm has evaluated the developer's performance. | **0.731** | 0.093 | 0.018 | 0.007 | −0.012 | −0.045 |
| There is consistency between efforts and rewards. | **0.711** | 0.040 | −0.037 | −0.060 | 0.123 | 0.195 |
| It is likely that the developer will be able to achieve the project objectives satisfactorily. | **0.625** | 0.075 | 0.181 | −0.011 | −0.034 | −0.340 |
| The developer has a good reputation in the marketplace. | 0.048 | **0.772** | −0.029 | 0.004 | −0.025 | 0.089 |
| Problems between the developer and the buyer are resolved through friendly negotiations. | 0.009 | **0.759** | 0.109 | 0.003 | 0.200 | −0.093 |
| The developer and the buyer share knowledge effectively whenever necessary. | 0.000 | **0.675** | 0.147 | 0.225 | 0.041 | 0.285 |
| The buyer has a long-term business relationship with the developer. | −0.069 | **0.636** | −0.007 | −0.033 | −0.106 | 0.217 |
| The developer shares the same values and behavior as the buyer. | 0.077 | **0.604** | −0.040 | 0.415 | 0.238 | −0.196 |
| The developer tries their best to fulfill their commitments. | 0.093 | **0.582** | −0.135 | 0.459 | 0.066 | −0.099 |
| The developer takes a friendly stance and protects the buyer's interests. | 0.086 | 0.048 | **0.802** | −0.100 | 0.047 | 0.102 |
| The developer has a strong sense of social responsibility and receives public praise. | 0.154 | 0.007 | **0.712** | −0.045 | −0.031 | −0.025 |
| The developer mobilizes all kinds of resources to maintain a good relationship with the buyer. | 0.023 | 0.016 | **0.640** | 0.136 | 0.254 | −0.068 |
| Interest and risks are shared fairly and reasonably between developer and buyer. | −0.036 | −0.030 | −0.028 | **0.809** | 0.002 | 0.183 |
| The developer and the buyer have the same status and the parties do not belittle each other. | −0.021 | 0.378 | 0.054 | **0.689** | −0.027 | −0.031 |
| The developer and the buyer have common goals throughout every phase of the project. | 0.025 | −0.087 | 0.205 | 0.094 | **0.777** | −0.036 |
| The developer uses advanced technology and has good management skills. | −0.006 | 0.330 | 0.029 | −0.103 | **0.695** | 0.042 |
| The developer and the buyer are informed about each other's needs thanks to effective communications. | −0.058 | 0.284 | 0.090 | 0.051 | −0.071 | **0.793** |
| The developer and the buyer communicate frequently. | 0.386 | −0.080 | −0.235 | 0.242 | 0.301 | **0.463** |

Bolded numbers represent significant factor loadings.

*Factor 4: Fair play*

Sharing the risks fairly and not belittling each other defines fair play. This equality in status also means justice, openness, and sharing. Wong and Cheung [16], Cheung et al. [58] and Yeung et al. [59] investigated openness, alignment of efforts, sense of unity, respect system, and appreciation as a proxy for fair play affecting the level of trust between home buyers and developers.

*Factor 5: Unity of purpose*

Home buyers are unanimous in their wishes about project completion within budget, on schedule, and in good quality. As a result, home buyers usually expect developers to use the latest construction technologies rather than antiquated, costly, and time-consuming technologies. Similarly, they expect developers to use advanced materials that are cost-effective, functional, durable, and easy to install and to maintain. They also expect developers to have efficient and state-of-the-art management skills that can help them to meet the required budget, schedule, and quality requirements in a project. These preferences are implicit in typical home buyer wishes. One can, therefore, conclude that there is unity of purpose between the home buyer and the developer when the developer uses advanced management practices and the latest materials/technologies. This unity of purpose is important to build trust between the home buyer and the developer. The importance of unity of purpose is also pointed out in the literature that deals with issues of trust (e.g., Karlsen et al. [19]; Khalfan et al. [22]).

*Factor 6: Effective Communication*

The trust between the home buyer and the developer is enhanced when the communication between these two parties is extensive, effective, and frequent. The importance of good communication was confirmed in several studies on trust (e.g., Wong and Cheung [16]; Tai et al. [17]; Jiang et al. [18]; Karlsen et al. [19]). In a study conducted by Baş Aras [51], it was reported that the first of the 20 priorities of home buyers was expressed by the statement "I care about the communication style of the developer I will buy from".

Overall, it can be stated that the mutual trust between home buyers and developers must rely on the six factors identified in factor analysis, but especially the top two factors with the largest factor loadings, namely integrity and close interaction. Integrity requires that home buyers and developers avoid taking advantage of each other, developers perform reliably of the construction site while home buyers make their installment payments in a timely manner, the home buyer and the developer negotiate a contract that is clear and fair to both parties. In addition, close, constant, and smooth interaction implies sharing information on a regular basis and solving problems by means of friendly negotiations rather than destructive confrontations.

## 6. Conclusions

The value of residential building construction projects constitutes an important percentage of the value of construction projects. Residential building construction projects appeal to all segments of the society and are the growth engine of the country's economy. The "sell-build" delivery system is preferred by many home buyers and developers. The transactions involved in some of the "sell-build" projects are successful, while some are not. Failed transactions are common in Turkey. This problem primarily affects home buyers and developers, and indirectly affects the economy. The effects of "the home buyer's knowledge about the home purchasing process" and "the trust between home buyers and developers" on the success of these transactions have not been examined in detail. This study was conducted to fill this gap. In this study, transaction success was defined in terms of cost, time, and quality and was evaluated relative to the level of knowledge of the buyer about the purchasing process and the mutual trust between the developer and the buyer. A survey was conducted of 250 home buyers and 70 developers. The data collected was subjected to statistical analysis.

The agreement between a home buyer and a developer involves a promise on the part of the developer to deliver a residential unit with specific attributes in a certain period in exchange for a certain amount of money, whereas the home buyer promises to make regular payments until the sum of money specified in the agreement is paid up. This process progresses well if the home buyer is well informed about the process and the home buyer and the developer trust each other implicitly. This study confirmed that the home buyer's knowledge about the home buying process and the trust between the home buyer and the developer affect the success of the transaction between home buyer and developer.

Home buyers come from different social and economic segments of the society and as a result, may have different levels of understanding about the construction process and the home buying experience. It is, therefore, essential that home buyers be fully informed about the issues involved in the home buying process. As per factor analysis, home buyers must particularly be informed about the terms of the contract, the developer's past performance and current performance, purchasing strategies, legal matters, and market conditions. Not only can informational materials be made available to them, but also training courses can be provided to potential home buyers by government agencies and non-governmental organizations.

Ensuring trust between home buyer and developer is extremely important. In sell-build projects, the home buyer and the developer build a long-term relationship through the design and construction phases that may last over a year and perhaps longer. In this long process, the home buyer needs to trust the developer as the buyer hands over his/her hard-earned savings to the developer in several installments. On the other hand, the developer also needs to trust the home buyer at every stage of the project to secure an uninterrupted financing inflow provided by the buyer. As per factor analysis, trust can be built if integrity, close interaction, harmony, fair play, unity of purpose, and effective communication are part of the equation.

This study's contribution to the transaction of home buying is its emphasis on the importance of the home buyer's knowledge about the home purchasing process and on the trust between the home buyer and the developer. While it is part of the developer's business to be knowledgeable about this kind of transaction, the home buyer should try to be well informed about the transaction by reading relevant materials, consulting specialists, and/or attending training courses. To prove trustworthiness, while the developer should perform well in present and past projects (i.e., on schedule, within budget, and top quality), the home buyer should be consistently punctual in making the installment payments.

The first limitation of this study is that it considers only two factors affecting the success of the home buying transaction, namely the knowledge of the home buyer about the transaction process, and the trust between home buyer and developer. Future studies can consider additional factors such as inflation and developer capacity. The second limitation involves the use of cost, time, and quality to measure the success of a transaction. Other criteria of success may involve work safety and contribution to the environment (e.g., energy efficient green home) and may be considered in future work.

**Author Contributions:** Conceptualization, S.A. and D.A.; methodology, S.A. and D.A.; software, S.A.; validation, S.A. and D.A.; formal analysis, S.A. and D.A.; investigation, S.A.; resources, S.A.; data curation, S.A. and D.A.; writing—original draft preparation, S.A.; writing—review and editing, S.A., D.A. and G.T.-Ç.; visualization, S.A., D.A. and G.T.-Ç.; supervision, D.A. and G.T.-Ç.; project administration, S.A., D.A. and G.T.-Ç.; funding acquisition, S.A. and G.T.-Ç. All authors have read and agreed to the published version of the manuscript.

**Funding:** This research was funded by the Scientific Research Project (BAP) unit of Çukurova University, Adana, Turkey with project number FDK-2019-11331.

**Institutional Review Board Statement:** Not applicable.

**Informed Consent Statement:** Not applicable.

**Data Availability Statement:** Data is contained within the article and Appendix C.

**Conflicts of Interest:** The authors declare no conflict of interest.

## Appendix A

**Table A1.** Survey Administered to Home Buyers.

| General Information Questions | | | | |
|---|---|---|---|---|
| Your gender? | | | | |
| Your age? | | | | |
| Your city of residence? | | | | |
| Your educational status? | | | | |
| Your occupation? | | | | |
| Monthly income? | | | | |
| In the trilogy of duration, cost and quality, which one comes first for you and which comes after? (Rank 1 to 3) | Duration (       ) | | | |
| | Cost (       ) | | | |
| | Quality (       ) | | | |
| Is your home purchasing transaction continuing (or is completed) successfully? | Duration | Successful (    ) | Unsuccessful (    ) | |
| | Cost | Successful (    ) | Unsuccessful (    ) | |
| | Quality | Successful (    ) | Unsuccessful (    ) | |

**Please indicate your opinion about each of the following statements concerning your knowledge of the home purchasing transaction**

| Statements concerning your knowledge of the home purchasing transaction | Strongly Disagree | Disagree | Neutral | Agree | Strongly Agree |
|---|---|---|---|---|---|
| 1. I researched the developer's finances and human/equipment resources. | | | | | |
| 2. I know the developer's project management capabilities. | | | | | |
| 3. I signed the contract after I read it thoroughly and understood its content. | | | | | |
| 4. The contract I signed contains all the construction related details I expect to see in such a contract. | | | | | |
| 5. I made a detailed comparison between the information specified in the blueprints and the information specified in the contract. | | | | | |
| 6. I am familiar with the brands, standards, and quality of materials used in the construction. | | | | | |
| 7. I know how much my home will cost at handover. | | | | | |
| 8. I can calculate the future value of the current money I invested in the building. | | | | | |
| 9. I know about the level of success and reputation of the developer in past projects. | | | | | |
| 10. After handover, I know how long it took the developer to build the superstructure and to complete the specialty work. | | | | | |
| 11. I compared the residential development project with alternative projects. | | | | | |
| 12. I talked to other buyers about the residential development project. | | | | | |
| 13. I researched the proposed landscaping around my potential home and the potential development of the neighborhood. | | | | | |

**Table A1.** *Cont.*

| Please indicate your opinion about each of the following statements concerning your knowledge of the home purchasing transaction | | | | | |
|---|---|---|---|---|---|
| **Statements concerning your knowledge of the home purchasing transaction** | **Strongly Disagree** | **Disagree** | **Neutral** | **Agree** | **Strongly Agree** |
| 14. I researched with whom I would be a neighbor. | | | | | |
| 15. I received support from technical experts and relevant lawyers who have information about residential projects. | | | | | |
| 16. I know the legal regulations about the construction process. | | | | | |
| 17. I made the decision to purchase my home rationally rather than emotionally. | | | | | |
| 18. In the construction process, I consider hiring a supervisor to follow the whole process. | | | | | |
| 19. I am familiar with dispute resolution methods. | | | | | |
| 20. I know construction terms such as title deed, property, easement, mortgage, and building permit. | | | | | |
| **Please indicate your opinion about each of the following statements concerning your trust in the developer** | | | | | |
| **Statements concerning your trust in the developer** | **Strongly Disagree** | **Disagree** | **Neutral** | **Agree** | **Strongly Agree** |
| 1. The developer uses advanced technology and has good management skills. | | | | | |
| 2. The developer behaves honestly in the construction process and in handling chage orders. | | | | | |
| 3. Problems between the developer and the buyer are resolved through friendly negotiations. | | | | | |
| 4. The developer shares the same values as the buyer. | | | | | |
| 5. The developer and the buyer share knowledge effectively whenever necessary. | | | | | |
| 6. The developer tries their best to fulfill their commitments. | | | | | |
| 7. The developer has a good reputation in the marketplace. | | | | | |
| 8. The developer and the home buyer do not belittle or disrespect each other. | | | | | |
| 9. The buyer has a long-term business relationship with the developer. | | | | | |
| 10. Interest and risks are shared fairly and reasonably between developer and buyer. | | | | | |
| 11. The developer and the buyer are informed about each other's needs thanks to effective communications. | | | | | |
| 12. The developer and the buyer communicate frequently. | | | | | |
| 13. There is consistency between efforts and rewards relative to implementation. | | | | | |
| 14. The developer acts reliably and is capable of meeting the buyer's expectations. | | | | | |
| 15. The buyer has confidence in the competence of the developer. | | | | | |

**Table A1.** *Cont.*

| **Please indicate your opinion about each of the following statements concerning your trust in the developer** | | | | | |
|---|---|---|---|---|---|
| **Statements concerning your trust in the developer** | **Strongly Disagree** | **Disagree** | **Neutral** | **Agree** | **Strongly Agree** |
| 16. The rights and obligations of the developer and the buyer are clearly expressed in the contract. | | | | | |
| 17. The developer does not take advantage of the weak points in the contract. | | | | | |
| 18. A consulting firm has evaluated the developer's performance. | | | | | |
| 19. It is likely that the developer will be able to achieve the project objectives satisfactorily. | | | | | |
| 20. The home buyer has successfully collaborated with the developer in this project. | | | | | |
| 21. The developer has a strong sense of social responsibility and receives public praise. | | | | | |
| 22. The developer takes a friendly stance and protects the buyer's interests. | | | | | |
| 23. The developer and the buyer have common goals throughout every phase of the project. | | | | | |
| 24. The developer mobilizes all kinds of resources to maintain a good relationship with the buyer. | | | | | |

## Appendix B

**Table A2.** Survey Administered to Developers.

| **General Information Questions** |
|---|
| Your educational status? |
| Your city of residence? |

| **Please indicate your opinion about each of the following statements concerning the home buyer's knowledge of the home purchasing transaction** | | | | | |
|---|---|---|---|---|---|
| **Statements concerning the home buyer's knowledge of the home purchasing transaction** | **Strongly Disagree** | **Disagree** | **Neutral** | **Agree** | **Strongly Agree** |
| 1. The buyer investigates the developer's finances and human/equipment resources. | | | | | |
| 2. The buyer knows the developer's project management capabilities. | | | | | |
| 3. The buyer signs the contract after he/she reads it thoroughly and understands its content. | | | | | |
| 4. The buyer knows all the details about the construction project in the signed contract. | | | | | |
| 5. The buyer makes a detailed comparison between the information specified in the blueprints and the information specified in the contract. | | | | | |
| 6. The buyer is familiar with the brands, standards and quality of materials used in the construction. | | | | | |

**Table A2.** *Cont.*

| **Please indicate your opinion about each of the following statements concerning the home buyer's knowledge of the home purchasing transaction** | | | | | |
|---|---|---|---|---|---|
| **Statements concerning the home buyer's knowledge of the home purchasing transaction** | **Strongly Disagree** | **Disagree** | **Neutral** | **Agree** | **Strongly Agree** |
| 7. The buyer knows how much his/her home will cost at handover. | | | | | |
| 8. The buyer can calculate the future value of the current money he/she invested in the project. | | | | | |
| 9. The buyer knows about the level of success and reputation of the developer in past projects. | | | | | |
| 10. After handover, the buyer knows how long it took the developer to build the superstructure and to complete the specialty work. | | | | | |
| 11. The buyer compared the residential development project he/she purchased with alternative projects. | | | | | |
| 12. The buyer talked to other buyers about the residential development project. | | | | | |
| 13. The buyer inquired about the proposed landscaping around his/her potential home and the potential development of the neighborhood. | | | | | |
| 14. The buyer inquired about his/her potential neighbors. | | | | | |
| 15. The buyer received support from technical experts and lawyers who specialize in residential projects. | | | | | |
| 16. The buyer knows the legal regulations about the construction process. | | | | | |
| 17. The buyer decided to purchase his/her home rationally rather than emotionally. | | | | | |
| 18. In the construction process, the buyer considered hiring a supervisor to monitor the whole process. | | | | | |
| 19. The buyer is familiar with dispute resolution methods. | | | | | |
| 20. The buyer knows construction terms such as title, deed, property, easement, mortgage, and building permit. | | | | | |
| **Please indicate your opinion about each of the following statements concerning your trust in the home buyer** | | | | | |
| **Statements concerning your trust in the home buyer** | **Strongly Disagree** | **Disagree** | **Neutral** | **Agree** | **Strongly Agree** |
| 1. The buyer is solvent and has sufficient knowledge about the construction market. | | | | | |
| 2. The buyer acts honestly in their relationship with the developer. | | | | | |
| 3. Problems between the developer and the buyer are resolved through friendly negotiations. | | | | | |
| 4. The buyer shares the same values as the developer. | | | | | |
| 5. The developer and the buyer share knowledge effectively whenever necessary. | | | | | |

**Table A2.** *Cont.*

| Please indicate your opinion about each of the following statements concerning your trust in the home buyer | | | | | |
|---|---|---|---|---|---|
| Statements concerning your trust in the home buyer | Strongly Disagree | Disagree | Neutral | Agree | Strongly Agree |
| 6. The buyer tries their best to fulfill their commitments. | | | | | |
| 7. The buyer comes from a reputable segment of the society. | | | | | |
| 8. The developer and the buyer do not belittle or disrespect each other. | | | | | |
| 9. The developer has a long-term business relationship with the buyer. | | | | | |
| 10. Interest and risks are shared fairly and reasonably between developer and buyer. | | | | | |
| 11. The developer and the buyer are informed about each other's needs thanks to effective communications. | | | | | |
| 12. The developer and the buyer communicate frequently. | | | | | |
| 13. There is consistency between efforts and rewards relative to implementation. | | | | | |
| 14. The buyer acts reliably and can meet the developer's expectations. | | | | | |
| 15. The developer has confidence in the competence of the buyer. | | | | | |
| 16. The rights and obligations of the developer and the buyer are clearly expressed in the contract. | | | | | |
| 17. The buyer does not take advantage of the weak points in the contract. | | | | | |
| 18. The developer regularly evaluates the payment performance of the buyer. | | | | | |
| 19. The buyer makes his/her payments in a timely manner. | | | | | |
| 20. The developer has collaborated successfully with the buyer in this project. | | | | | |
| 21. The buyer has a strong sense of social responsibility. | | | | | |
| 22. The buyer takes a friendly stance and protects the developer's interests. | | | | | |
| 23. The developer and the buyer have common goals throughout every phase of the project. | | | | | |
| 24. The buyer mobilizes all kinds of resources to maintain a good relationship with the developer. | | | | | |

# Appendix C

**Table A3.** Descriptive Statistics.

| Survey Administered to Home Buyers | | | | Survey Administered to Developers | | | |
|---|---|---|---|---|---|---|---|
| Statements Related to the Buyer's Knowledge about the Transaction | Mean Scores * | Statements Related to Buyer's Trust in Developer | Mean Scores * | Statements Related to the Buyer's Knowledge about the Transaction | Mean Scores * | Statements Related to Developer's Trust in Buyer | Mean Scores * |
| 1 | 3.82 | 1 | 3.58 | 1 | 3.44 | 1 | 3.64 |
| 2 | 3.77 | 2 | 3.42 | 2 | 3.51 | 2 | 3.51 |
| 3 | 3.98 | 3 | 3.44 | 3 | 3.76 | 3 | 3.79 |
| 4 | 3.74 | 4 | 3.43 | 4 | 3.63 | 4 | 3.17 |
| 5 | 3.91 | 5 | 3.56 | 5 | 3.86 | 5 | 3.30 |
| 6 | 3.59 | 6 | 3.48 | 6 | 3.16 | 6 | 3.37 |
| 7 | 3.84 | 7 | 3.42 | 7 | 3.57 | 7 | 3.10 |
| 8 | 3.50 | 8 | 3.39 | 8 | 3.23 | 8 | 3.50 |
| 9 | 3.69 | 9 | 3.10 | 9 | 3.60 | 9 | 3.13 |
| 10 | 3.32 | 10 | 3.20 | 10 | 3.66 | 10 | 3.57 |
| 11 | 3.88 | 11 | 3.32 | 11 | 3.79 | 11 | 3.26 |
| 12 | 3.67 | 12 | 3.17 | 12 | 3.47 | 12 | 3.81 |
| 13 | 3.61 | 13 | 3.19 | 13 | 4.04 | 13 | 3.86 |
| 14 | 3.54 | 14 | 3.32 | 14 | 3.33 | 14 | 3.51 |
| 15 | 3.73 | 15 | 3.45 | 15 | 3.09 | 15 | 3.23 |
| 16 | 3.61 | 16 | 3.43 | 16 | 3.26 | 16 | 4.34 |
| 17 | 3.67 | 17 | 3.34 | 17 | 3.49 | 17 | 2.97 |
| 18 | 3.61 | 18 | 3.39 | 18 | 2.49 | 18 | 3.69 |
| 19 | 3.80 | 19 | 3.36 | 19 | 3.00 | 19 | 3.71 |
| 20 | 3.90 | 20 | 3.46 | 20 | 3.46 | 20 | 3.86 |
| | | 21 | 3.44 | | | 21 | 3.30 |
| | | 22 | 3.56 | | | 22 | 3.19 |
| | | 23 | 3.61 | | | 23 | 3.81 |
| | | 24 | 3.28 | | | 24 | 3.36 |

* Means calculated on a scale of 1–5, with 1 = strongly disagree, and 5 = strongly agree.

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
