# Peer review of "Factors That Affect the Level of Success of the Transaction between Home Buyers and Developers in Sell-Build Residential Projects"

_buildings, doi:10.3390/buildings11030127_

Round 1

Reviewer 1 Report

Overall, the paper is well-structured and each step is elaborated in sufficient detailThe topic is very relevant in the residential construction projects. There are only few comments on the paper.

  • In section 3.1, it is mentioned that 20 questions were included in the survey. But, in table 2, 13 knowledge items are presented. It is better to provide more information on each of these items. Maybe some constructs are measured using multi-item measurement scale.
  • In section 4, some findings were mentioned about the demographics of the respondents. But no explanations were presented on how the survey were administrated. In other words, how the participants were approached for this survey? Is it a random selection of the participants?
  • In section 4.1 and 4.2, you can mention that your analysis met the thresholds of the factor analysis by referring it to the book “Multivariate data analysis: a global perspective” by Hair et. al.
  • The aim of the exploratory factor analysis  performed in sections 4.1 and 4.2 is not clear. Why these factors are extracted? Maybe the effect of these factors on transaction success can be explored to see which of these factors affect the level of success more significantly.

Reviewer 2 Report

Overall I enjoyed reading the article - it presents a well written and effective analysis of some interesting results of two surveys capturing the two supply and demand sides of residential development in Turkey. There are some minor improvement that I would suggest as follows:

  1. include the term 'Sell-build' in the title
  2. include Turkey in keywords
  3. line 52 - victimisation is a an emotive term - better to frame in more sober legal terminology
  4. Section 2.2  - is sell-build more common in Turkey than other countries (in Europe)? Offer some comparison and contextualisation; may be cultural differences; states later on in article that more failures - present the evidence of this here rather than at end
  5. Line 150-151 - the point being made is unclear; please explain.
  6. Section 2.3  - replace listing of day to day jobs/roles with simple statement that most purchasers are 'lay' e.g. non-expert - the important distinguishing feature is that they are not informed clients (use this term elsewhere in article)
  7. Line 166-167 - need to introduce law of contract and tort and legal principle of 'caveat emptor' or buyer beware; should be both statute and case law to support this
  8. Table 1  - I am happy with hypotheses 1,2 and 3 however less so with framing of 4 and 5 which vary between how they are presented in this table and how they are articulated on page 9 e.g. Hypothesis 4 relates to developer's perception of the knowledge of purchasers e.g. nuanced and subjective (perceptions). With regard Hypothesis 5 this is not a simple/crude oppositional developer versus buyer relationship and needs to be framed/expressed differently.
  9. Section 4 - need to compare with national profile for Turkey to demonstrate whether representative or not
  10. Line 303 H1 quite knowledgeable = informed client
  11. Line 337 H4  - introduce term mis-match to capture different perceptions; also see point 8
  12. Table 5 see point 8 about re-framing of H5
  13. Line 379 make this point earlier on in article as affects comparison with other countries/jurisdictions - links to earlier point about law of contract
  14. Section 5 Conclusions - term 'locomotive role' does not work; suggest replaces with growth engine or multiplier effect; mention made of high number of failed transaction in Turkey - this needs introducing earlier on as part of context for study supported by evidence 

Reviewer 3 Report

It's an interesting article.

About the bibliographical references in  par. 2.2 and 2.3, it would be interesting to investigate the limits of the mentioned researches, and the aspects that your research helps to improve.

About the variables that influence the negotiation of properties see also:

Semeraro P. and Fregonara E. (2013), The impact of house characteristics on the bargaining outcome. In International Journal of European Real Estate Research, vol.6 (3),  pp.262-278.

Morano P., Tajani F and Locurcio M. (2018), Multicriteria analysis and genetic algorithms for mass appraisal in Italian property market. In International Journal of Housing Markets and Analysis, vol.1, pp.229-262.

Reviewer 4 Report

Using survey data, the paper examines the effect (1) of homebuyers’ knowledge of the construction process and (2) of homebuyers-developers reciprocal trust, on the level of success of sell-build projects in Turkey. The results come at no surprise to corroborate what is already known in other markers: better knowledge and trust between the parties are factors of success. Overall the paper is well written and well structured and I think it can be published once it resolves the issues I raise below.

My main concern is about the basic variable the paper uses (the transaction success - OST). In my view it is not clear in the paper how this variable has been constructed. It is said that the OST is measured as a combination of the overall cost, time and quality of the project (page 6, line 244) implying that projects of higher cost, time of completion and of lower quality compared to a datum (i.e. a point of reference) are considered to be less successful. Yet, it is not clear what this datum is. Is it the average values of cost/time/quality for the sample, of similar projects? Is it the cost/time/quality figures as perceived by the buyer? Is it the values of cost/time/quality that were initially agreed between the buyer-developer? or those figures specified in the contract?

Another point of concern is the interpretation provided with regard to hypothesis 4 (page 9, lines 339-344). The paper finds “a significant difference between homebuyer’s knowledge of the purchasing process and the developer’s perception of this knowledge”. This, according to the paper, means that buyers’ knowledge is “not as realistic” (line 343) or is “unrealistically inflated” (line 340), putting the onus exclusively on the homebuyers. I have a different understanding of the finding. In my view It could mean that buyers' knowledge is better in comparison to what developers think of if, or that developers underestimate homebuyers’ real knowledge.

Of lower importance are the following points (ranked below not in terms of importance but as they appear in the text) which also need to be addressed before publication

  • The literature review provided in pages 3-4 (lines 117-149) needs to be rewritten in order to reflect the discussion on the basis of the factor examined and not of each individual paper.
  • The items in Table 2 (pages 5-6, lines 226-227) should be accompanied with a short explanation of what each item is about (e.g. what ‘selection’ or ‘responsibility’ means in this context).
  • Similarly, the items that reflect trust (page 6, lines 232-236) should be adequately explained. A table can be used to summarise this information.
  • The last three rows of Table 3 (page 8) need to be explained: what they mean, how they have been calculated, and how the weights used in eq1 come up.
  • Lines 309-310 write “…transactions that may not be successful.”The paper should make clear what ‘not successful’ means in this context.
  • Table 5 (page 10, lines 349-350) should also provide the figures or the levels of statistical significance.
  • In Table 6 (page 11) it is not clear how the homebuyers statement “I can calculate the future value of the current money I invested in the building” is related to the presumed factor “developers’ past and present performance”.
  • Similarly, in Table 7 (page 13) it is not clear how the homebuyers statement “The developer uses advanced technology and has good management skills” is related to the presumed factor “unity of purpose”.
  • A copy of the questionnaires used in the study should be provided (in an appendix), as well as the outputs of the statistics performed

Round 2

Reviewer 4 Report

The revised draft of the paper dealt with some of the issues I raised in the first draft, but not adequately. Especially the first point I made, which I think is extremely important (repeated again below) has not been dealt at all by the authors. Moreover, the paper does not add much to what the literature already knows (which is: better knowledge and trust between the parties are factors of success in a transaction). On these grounds my recommendation is for the paper to be rejected from publication.

Below I elaborate more on the problems I see with the draft provided.  

The main point I raised in the first draft of the paper has not been dealt at all by the authors. I repeat it here again.

My main concern is about the basic variable the paper uses (the transaction success - OST). In my view it is not clear in the paper how this variable has been constructed. It is said that the OST is measured as a combination of the overall cost, time and quality of the project (page 6, line 244) implying that projects of higher cost, time of completion and of lower quality compared to a datum (i.e. a point of reference) are considered to be less successful. Yet, it is not clear what this datum is. Is it the average values of cost/time/quality for the sample, of similar projects? Is it the cost/time/quality figures as perceived by the buyer? Is it the values of cost/time/quality that were initially agreed between the buyer-developer? or those figures specified in the contract?

The statement provided in lines 372-377 does not really answer the concern I raised in the 1st draft: “In my view the ‘a significant difference between homebuyer’s knowledge of the purchasing process and the developer’s perception of this knowledge’ could mean that buyers' knowledge is better in comparison to what developers think of if, or that developers underestimate homebuyers’ real knowledge.”

In addition, the paper still needs to take into account the following points I already raised in the 1st draft but not considered adequately

  • The literature review provided in pages 3-4 (lines 117-149) needs to be rewritten in order to reflect the discussion on the basis of the factor examined and not of each individual paper.
  • The items that reflect trust (3.2 section) should be explained ADEQUATELY. A table to be used to summarise this information.
  • The last three rows of Table 3 (page 8) need to be explained:
    • what they mean, what “successful” and “unsuccessful” mean with regard to duration/cost/quality in the success of the transaction?
    • How they have been calculated; Responders marked them hierarchically? …a scale was used? …something else?
    • How the weights come up?
  • Lines 336-337 (revised draft) write “…transactions that are considered "unsuccessful" by respondents in at least one of the success measures of time, cost, or quality.”The paper should clarify what this means.
  • The explanation provided in Table 6 (page 11) regarding how the statement “I can calculate the future value of the current money I invested in the building” is related to the presumed factor “developers’ past and present performance” is still problematic. On what grounds someone’s ability to calculate the future value of his own investment is a measure of someone’s else performance? And second, the evaluation of a company’s performance is not simply a matter of knowing how to calculate the future value of money. If only things were so simple…
  • Similarly, in Table 7 (page 13) it is not clear how the homebuyers statement “The developer uses advanced technology and has good management skills” is related to the presumed factor “unity of purpose”. Why someone’s perception that the second party in the transaction uses advanced technology indicates that there is a unity of purpose? It could equally mean exactly the opposite.
  • A copy of the questionnaires used in the study HAS TO BE PROVIDED (in an appendix), as well as the outputs of the statistics performed

Round 3

Reviewer 4 Report

The revised draft is an improvement that advances the quality of the paper. Issues that were “blatantly obvious” to the authors are now much clearer to the readers too, who are the target of the journal.

Two minor issues need to be resolved before publication.

  1. Still it is not crystal clear how OST was measured. I reckon the text needs to be more precise in lines 250-257, where the OST appears. Perhaps a wording close to the follwoing would remove any doubts

“As per the consensus in the literature (see Section 2.1), transaction success was measured by project actual cost vs. initially budgeted, actual completion time vs. initially scheduled, and actual quality vs. initially specified.

  1. The appendix should provide the questionnaire, plus the statistics, which have not been provided in the last draft. The provision of the questionnaire would help to clarify most of the issues raised in this review process, to the benefit of the readers
